# A Transformer-Based Machine Learning Approach for Sustainable E-Waste Management: A Comparative Policy Analysis between the Swiss and Canadian Systems

Saidia Ali and Farid Shirazi *

Ted Rogers School of Information Technology Management, Toronto Metropolitan University, Toronto, ON M5B 2K3, Canada
* Correspondence: f2shiraz@ryerson.ca

**Abstract:** Efficient e-waste management is crucial to successfully achieve sustainable urban growth universally. The upsurge in e-waste has resulted in countries, including Canada, adopting a wide array of policies associated with sustainable management. In this study, we conducted a mixed-method analysis of Canadian e-waste management policies to showcase the opportunities and limitations of the current system. We examine and compare the effectiveness of electronic waste management strategies in Canada and Switzerland using a comparative policy evaluation and by quantitatively measuring their efficiencies through two efficiency methods, namely a transformer-based, bidirectional, unsupervised machine learning model for natural language processing (NLP) and data envelopment analysis (DEA). Switzerland is utilized as a comparison case due to its robust legal framework that has been in place for proper management e-waste in order to enhance Canada's electronic waste management system. The policy considerations presented in this study are directed toward urban planners, policy makers, and corporate strategists. These involve a mix of political, economic, social, and environmental planning tools concerning how to communicate and foster competent e-waste management in these countries. This is the first study to incorporate DEA and NLP-based BERT analysis to identify the most efficient policy deployment concerning e-waste management.

**Keywords:** e-waste; sustainability; extended producer responsibility; recycler qualification program; $CO_2$ emission; data envelopment analysis; natural language processing; machine learning; BERT

## 1. Introduction

In the realms of urban sociology and urban planning, the ongoing debate amongst academics has dealt with the various ways to tackle the plethora of environmental issues faced by the world in recent years. A common solution that many scholars, environmentalists, policymakers, and urban planners agree upon is sustainable development, which has largely been seen as a legally unenforceable resolution [1,2]. In recent years, many academics [3–6] have researched the origins and meaning of sustainability. However, the recent popularity of the word sustainability, specifically urban sustainability and green cities, should require us to rethink and redefine sustainability as an idea that will allow us to examine, explain, and critique current urban sprawl, which may distort the economic, social, political, and cultural processes in the widespread execution of urban development [7]. The concept of sustainability is not only rooted in the various avenues of human activities but also in the various kinds of social structures, such as a city whose urbanization is mainly founded upon its environmental, social, and political surroundings [8]. As world population and mass consumption continue to trend upward, urban development have consequences in urban cities [8]. Hence, it is crucial to utilize the concept of sustainability for the positive expansion of cities to protect their ecosystems [8].

A major area of concern for Canada and other countries is the absence of sustainable strategies and policies for producing, using, and managing electronic waste [9]. Electronic waste (commonly referred to as e-waste) and waste electrical and electronic equipment (WEEE) is utilized globally and incorporated into non-electronic items, such as stoves, washing machines, printers, and refrigerators. Electronic equipment has become an important aspect of contemporary societies by enhancing living standards; therefore, it is not surprising that the IT industry is a crucial element of individual countries, as well as national economies [10]. The rapid increase and flow of information technology devices in association with overall economic development has led to an overwhelming production of electronic waste because, as the production and use of such devices are attractive to the average consumer [11]. For example, in 2013, the average use of cell phones in Latin America and the Caribbean was 114.5 percent of the population, whereas broadband usage increased to 24 percent. Additionally, the Latin continents have experienced increased Internet consumption, with users comprising 46.7 percent of the population [12].

The increased expansion and consumption of electronic and electrical equipment (EEE) also points to the vast amounts of waste that countries have produced. It is estimated that the total weight (not including photovoltaic panels) of electronic waste consumption worldwide has increased by 2.5 million metric tons [11]. Moreover, in 2019, approximately 53.6 million metric tones of electronic waste were produced globally, equal to 7.3 kg per capita [11]. Since 2014, the production of electronic waste has increased by 9.2 metric tons, and it is estimated that by 2030, this number will reach 74.7 metric tons [11]. The considerable increase in electronic waste generation is due to increased consumption rates, shorter life cycles of electronics, and reduced avenues for repair and maintenance [11]. In 2019, Asia produced the most electronic waste, at 24.9 metric tons, with the Americas next in line, producing 13.1 metric tons [11, followed by Europe, which generated 12 metric tons, whereas Oceania and Africa produced 0.7 and 2.9 metric tons, respectively [11]. Nationally, Europe topped the list for the production of electronic waste on a per capita basis, generating 16.2 kg per capita [11], followed by Oceania (16.1 kg per capita), the Americas (13.3 kg per capita), Asia (5.6 kg per capita), and Africa (2.5 kg per capita) [11]. It is not easy to calculate the amount of electronic waste produced by developing nations because there are discrepancies between the various quantifications utilized by stakeholders [13]. In addition, the regulatory and management systems in third-world countries are underdeveloped and even non-existent in some places, so electronic waste management often occurs through the informal sector under precarious conditions [11].

In the past twenty years, politicians, policy makers, recyclers, producers, and various stakeholders have created specialized treatment and recovery facilities in some nations that are responsible for collecting and treating e-waste from owners suitable only for those instances. However, with many of these facilities put into place, there is still an absence of reporting, monitoring, and evaluation of e-waste consumption and production data, owing to the lack of management strategies. In most countries, e-waste is not collected and is disposed of with improper techniques, so components and are not recovered, with waste often shipped to underdeveloped nations, further increasing the waste management issue [14].

By 2014, the total number of countries that have embraced a national electronic waste management policy, legislation, or regulatory scheme had risen to 78 from 61 [11]. However, the implementation of such policy measures in these countries has been slow, coupled with poor enforcement, monitoring, and evaluation due to an absence of economic investments and political will [11]. Owing to the slow adoption of policy measures to effectively manage electronic waste, the repercussions, such as increased greenhouse gases, depletion of resources, and the release of harmful chemicals during informal recycling practices, have showcased the issue of staying within sustainable boundaries [11]. Even nations with formal electronic waste management systems are challenged with disposal and recycling rates. Moreover, the product scope incorporated in policies varies considerably depending on the electronic classification systems that are commonly utilized. These differences in

product scopes result in the absence or lack of organization of electronic waste across nations [11,15]. Hence, the enforcement of policies coupled with monitoring methods for tracking the flow of electronic waste within and across nations is crucial for sustainable and circular economies [11,16].

Canada is one a wealthy country that is also confronted with the challenge of effectively implementing, evaluating, and monitoring policies associated with the recycling, reduction, and reuse of electronic waste while encouraging producers to manufacture safer and eco-friendly electronics [17–19]. Provincial and federal governments in Canada have attempted to implement regulatory tools for the management of electronic waste; however, a majority of the electronic waste management systems in Canada have underperformed [19]. Through an action framework, McKerlie et al. [20] examined Canada's enforcement of the extended producer responsibility (EPR) framework. The responsibility for waste management in Canada is primarily distributed between various levels of government. There is no effective federal electronic waste legislation for electronic waste management in Canada, but many stewardship programs exist in Alberta, Ontario, and Nova Scotia [20]. Moreover, jurisdictional boundaries have affected waste management activities in Canada, in addition to the fact that Canadian municipalities having a limited range of resources and funding to reduce and manage their electronic waste generation [19].

*Importance of Selecting Switzerland*

As mentioned previously, one of the main research goals of this study is to examine the various electronic waste policies and regulations that have been successfully established in another nation as a reference point for the betterment of Canada's current electronic waste management regulations. Thus, Switzerland was chosen as the most suitable country for comparison because environmental sustainability problems are an important aspect for the Swiss government and its residents [21]. In 2018, Switzerland was ranked first on the Environmental Sustainability Index [21]. Moreover, according to the 2022 EPI report [22], Switzerland is the top-raking country on a global scale when it comes to managing solid household fuels, reducing nitrogen and sulfur dioxide growth rates, solid waste, and waste management.

Moreover, the country has been a worldwide leader in setting standards through its waste management practices, making commendable contributions in the area, showcasing their government's ability to be transparent, interregional cooperation, information policy, and public participation [23]. Furthermore, Switzerland has been the first and most successful country in the waste management field for the past twenty years, even outperforming countries with the most protuberant electronic waste management legislation, such as the EU WEEE Directive, which was presented in [21]. Finally, the Swiss electronic waste management system functions through a state-of-the-art process comprising a system based upon the most advanced technical procedures [21]. The developed system in Switzerland has proven to be very beneficial, as it has allowed the country to use such innovative developments in other facilities, thus proving its economically viability [21]. It is important to remember that the goal of this thesis is not to recommend a one-size-fits-all approach. Here, we use Switzerland as a reference to discuss some essential policy options on a key issue to provide vital insights with respect to applying an EPR-based e-waste management system in Canada.

Moreover, there is an absence of scholarly work examining Canadian policies of effective electronic waste management and how the current Canadian system can be improved by creating a clear framework that combines e-waste management and urban sustainability [24,25].

Our main research questions for this study are:

1.  What are the challenges and opportunities in Canada's current e-waste management system?
2.  What policy interventions can be used to improve the Canadian e-waste management system and possibly overcome the observed challenges?
3.  What lessons can Canada adopt from the Swiss e-waste management system?

## 2. Related Works

The literature review is divided into two sections: the sustainable development literature and the electronic waste literature. We will begin our review of the sustainable development literature by providing an explanation of the term sustainability. The importance of implementing a sustainable waste management system for urban cities will also be explained by focusing on the importance of social, economic, and environmental sustainability in urban areas.

In modern times, the word sustainability has become prevalent, especially in political agendas, business models, and various strategic documents, such as the European Union's 2007 Lisbon Treaty [26].

In written records, the complete definitions pertaining to sustainable development and sustainability differ, although the terms have similar essential bases [27]. The term *sustainability* often refers to a collective system of quality, with its main principle placing healthy individuals at the center of a productive and harmonious life, as stated in *Article 1* of the Rio Declaration [28]. This principle is composed of the following three core facts. The 2030 Agenda for Sustainable Development, which was adopted by United Nations in 2015, provides 17 Sustainable Development Goals (SDGs), which are an urgent call for action by all countries—both developed and developing—in a global partnership [29] for sustainable development considering economic, social, and environmental strategies to improve health and education, reduce inequality, and spur economic growth—all while tackling climate change and working to preserve our oceans and forests [30].

In this context, as Moldan et al. [26] mentioned, the first fact of sustainable development emphasizes the welfare of individuals and their needs.

The second fact of sustainable development is that individuals should be healthy and live in synchronization with their natural surroundings [26]. Such a fact is important because it allows humans to successfully balance all three pillars of sustainability. Moreover, it suggests that individuals are social beings and are part of a holistic phenomenon dependent on complex associations—largely environmental relationships [26].

The last fact of sustainable development is that it is long-term [31]. This means that sustainable development considers past, present, and forthcoming generations and points to the idea of evolving circumstances of the future without any predestined timeline other than human life [26].

### 2.1. E-Waste Management

During the 1970s and 1980s, there was a common practice of transporting hazardous waste (including electronic waste) from developed countries to developing regions, such as Africa, Asia, and Central America [32]. This was practiced in Canada until the Philippines and Malaysia forced Canada to take back their contaminated waste in 2019 [33]. The export of hazardous waste was part of the "Not in My Backyard" movement in wealthier nations as citizens responded to anger and frustration with their government's inefficient handling of hazardous waste, including electronic waste [34]. The public's stand against this management in the 1970s led to stringent regulations in industrialized nations, such as the Resource Conservation and Recovery Act (RCRA), which was created in the United States of America in 1976 [32]. These laws resulted in inflation of the cost of eliminating hazardous waste in developed nations and deflation in developing nations [35,36].

There was also a growing concern that electronic waste would result in environmental issues, primarily when their waste management laws related to recycling, incineration, and landfilling were not implemented, in addition to a lack of environmental awareness among citizens and administrations [32]. In order to prohibit the "toxic trade", the Basel Convention came into effect in 1992 to strictly control the shipment of toxic waste from wealthier nations to developing nations [34]. The Basel Action Network defines *e-waste* as discarded appliances, from large household devices, cell phones, and personal computers to consumer electronics or, according to the OECD, any appliance using an electric power supply that has reached the end of its life [10].

Furthermore, a considerable proportion of disposed e-waste contains many valuable materials. The valuable materials found within electronic waste largely depend on the type and age of the waste [37]. An example is mobile phones and IT and telecommunications apparatus. Cell phones are made up of more valuable metals than ordinary household equipment [38,39]. Some important metals in cell phones include copper and tin, as well as rare metals, such as lithium, cobalt, antimony, silver, gold, palladium, etc. Specifically, 23 percent of the weight of a typical cell phone consists of valuable metals, with the rest being made of plastic and ceramic [37].

Moreover, many of the precious metals that are found within cell phones are also discovered within end-of-life (EOL) cathode ray tubes (CRTs), mainly in televisions and desktop monitors [10]. However, the problem is that many CRTs are coated with leaded glass to protect them from the x-rays projected on CRTs [37]. Therefore, countries such as Japan, some EU countries, and the USA have prohibited the disposal of CRTs in landfills because of their high toxicity levels [10].

In 2017, the Global E-Waste Monitor report stated that the quantities of raw and valuable compounds found within e-waste in 2016 were estimated to be worth USD 60 billion. The issue is that only a small percentage of this amount was extracted through electronic waste management techniques [40].

The main objective of electronic waste recycling has long been separation and resource recovery of the various raw materials contained within the waste [41]. Of all the metals embedded with electronic waste products, gold and copper have been extracted extensively due to their high financial value. However, other metals are also extracted when the methods utilized for separation and resource recovery are economically beneficial and practicable [42]. Therefore, chemical exposure needs to be considered [42]. For example, electronic devices are comprised of various plastics that may contain hazardous chemicals; hence, plastic contents within the e-waste must also be separated for proper recovery [43].

The most critical aspect of creating a zero-waste city is shifting from a linear economic model to a circular economy [44]. A 2019 study by the World Economic Forum estimated that only 9% of the global economy is circular, meaning that only 9% of items are reused or recycled into products [45]. The other 91% of the economy continues to follow a linear model of making and taking waste [46]. Moreover, it is crucial to demonstrate that collective strategies and regulations are needed to create efficient and sustainable management systems because they are a symbolic representation of a city's presentation. The transition toward a circular economy is impossible without a fundamental change in consumer behaviors regarding green purchase adaptation, new business models, and initiatives such as information campaigns, economic incentives, and strict regulations [47]. In this context, small- and medium-sized enterprises can play a major role, particularly in emerging countries. As mentioned in [48], all most countries, e-waste recycling and management are market-based activities; for example, in China, India, and South Africa, the e-waste management scheme is based on small- to medium-sized enterprises (SMEs), and each country is trying to overcome shortcomings in the current system by developing strategies to improve the system and encourage SMEs to participate in such activities.

Finally, it is essential to highlight the impact of illegal waste on environmental pollution and health. Illegal waste, including uncontrolled disposal of toxic and industrial material, landfilling, and unauthorized incineration, exerts unpredictable socioeconomic and public health damage [48]. In particular, during the landfilling of waste, a wide range of pollutants can be released into the environment. The existence of illegal landfills is an increasing global problem. Illegal, uncontrolled waste dumps occur most frequently on forest margins, in ditches, on the peripheries of inhabited areas, and other places ([49], p. 89).

## 2.2. The Importance of E-Waste Treatment

The treatment, disposal, and collection of electronic waste are important for countries to effectively create a sustainable management system. As mentioned in [50], it is estimated that global e-waste generation will increase from 53.6 million metric tons in 2019

to 74.7 million metric tons in 2030, an indicator that requires attention from policymakers, environmental managers, industries, and scientists with respect to the next cycle of e-waste generation.

As mentioned by Mudd [51], the mineral resources found in electronic devices are widely interpreted as finite or non-renewable. Although e-waste poses a massive problem for countries all across the globe, it should not be surprising to state that it is a golden opportunity for global economies. There is no doubt that e-waste is of considerable financial value due to the amounts of valuable substances found within each piece of WEEE (waste electrical and electronic equipment), such as silver, gold, platinum, palladium, etc. [52]. To put this into perspective, a typical smartphone contains 100 times more gold than a ton of gold ore [52]. Moreover, as most WEEE is disposed into landfills, these sites are a goldmine of valuable materials; therefore, more action must be taken to retrieve these resources [52]. However, according to Baldé et al. [41], only 20 percent of e-waste produced internationally is handled by formal recycling sectors. In developing countries, where populations largely consist of low- and middle-income earners, the greatest percentage of e-waste is handled by informal sectors with unsafe safety measures coupled with poor environmental conditions [53].

Gautam et al. [54] provide a detailed list of valuable base and precious metals extracted from electronic waste used in various industries. As the authors mentioned, a wide range of value-added products, such as high-purity base and precious metals, metal and metal oxide nanoparticles, nanostructured alloys, nanocomposites, microparticles, and composites, has been recovered from retrieved low-cost, ubiquitous electronic scrap [54]. Jaffe et al. [55] reported that a number of rare and energy-critical elements (ECEs) are currently critical to one or more technologies:

- Lithium (Li) is used in batteries for electric vehicles, cell phones, and laptops;
- Platinum-group elements are required constituents of fuel cells and could be used in other advanced vehicle applications;
- Silicon, indium, and tellurium are essential components of solar photovoltaic (PV) panels;
- Neodymium (Nd) and praseodymium (Pr) are used in wind turbines;
- Terbium (Tb) and europium (Eu) are used for lighting and displays;
- Rhenium (Re) is used in advanced high-performance gas turbines; and
- Helium (He) is used in cryogenics and many research applications. (p. 33, see [56] for other related ECE materials).

Most developed nations have implemented legislation, directives, and conventions for the adequate collection and treatment of electronic waste and its non-recyclable components [53]. These methods include the 3Rs (reduce, reuse, and recycle), extended producer responsibility (EPR), product stewardship, landfilling, incineration, etc. [10]. The European Union has already enforced two directives to force electronic waste producers to take back their devices so that the amount of waste going to landfills is condensed [57]. However, these systems do not exist in developing countries. Most electronic waste in developing nations is treated through backyard operations, such as through sky incineration, chemical leaching, and the use of smelters to recover various metals, such as gold, copper, and silver, with reduced yields and a majority of the waste being disposed at dumping sites, water reservoirs, and poorly constructed landfills, resulting in a plethora of environmental and health consequences [58].

Finally, electronic products need to be manufactured robustly so that they can be reused by consumers [51]. Specifically, products should be lighter and smaller with enhanced digitalization and cloud computing services. It is important to remember that many multinational corporations have pledged to reduce the number of wasteful components being utilized in their electronic supply chains. In contrast, other companies are committing to building electronics free of poisonous materials. These goals demand collaboration throughout the sector [51]. By creating better electronics designs, these products will be dispersed for longer timeframes and have prolonged lifespans, leading to their reuse and

refurbishment. Moreover, designing durable EEE through holistic methods will ultimately create greater value in a circular economy.

## 3. Methodology

The most challenging step in drafting research for comparative policy analysis is deciding on the measures that need to be evaluated. Many experts have suggested that the characterization of policy has been poorly constructed, with a lack of clearly defined concepts in past analyses. This has led to discrepancies in measurements, which have ultimately resulted in a plateau in terms of advancing explanations. Moreover, researchers have argued that more accuracy and emphasis are required to operationalize the dependent variable [59].

When researchers participate in comparative policy analysis, their points of comparison are not focused on a single piece of the policy at hand; instead, attention is placed on several features of public policy [60]. Scholars examining policy procedures often compare and contrast the various ways in which an issue is framed and brought to light in front of policy makers, as well as the pathways by which officials choose, devise, and enforce their policy objectives [61].

It is also important to compare and contrast policy quality, which entails ranking a set of policies based on specific assessment criteria, such as durability or coherence [62]. Policy change is another object that largely discusses and explains the changes in particular policy targets and means over a period, primarily using longitudinal methods to compare and contrast novel or tweaked policies against a previous goal [63]. The last example of an object in comparative policy analysis is related to policy outcomes, which tend to concentrate on the costs associated with political decisions to bring to light inefficient governmental responses, as well as actions that have led to positive results and can be adopted in other locations [64].

In this context, public policies comprise diverse central features that can be used for comparisons, such as objectives, targets, instruments, and agents [65]. Objectives focus on what a set of policies aspires to accomplish on a general scale and incorporate the finer details on the anticipated results linked with particular operative actions or attitudes and conditions that need to be transformed to address the issue [60]. This is the main method deployed in the present study. Moreover, concerning instrumental objectives—those directed toward devising a solution for an issue—policies frequently relate to goals, such as security, efficacy, and fairness [66]. In many cases, political records are written with a statement of the issue that needs to be resolved and the anticipated externalities of the policy methods; however, the goals of a policy intervention may be indirect and may therefore need to be probed through official documents or communication with political authorities [67].

### 3.1. Data Sources and Methods

For this study, we collected two sets of data. The first dataset is related to public documents associated with e-waste management in Canada and Switzerland. These documents include governmental reports; household surveys; provincial legislation, such as e-waste laws, amendments, and acts; annual and technical reports; white papers; and local research reports from environmental organizations, such as SWICO, SENS, EPR Canada, and Call2Recycle Canada (see Table A1 in Appendix A for more detail).

The information in Table A1 (Appendix A) demonstrates the crucial legislative drivers of e-waste laws and the institutional makeup in Canada and Switzerland. Moreover, the table showcases the differences and similarities regarding the institutional framework and variations in the development of e-waste management approaches embraced by the various markets that comprise the EEE industry. This is a considerable hurdle, especially in the Canadian context due to jurisdictional differences and the wide range and complexity of WEEE and materials covered across provinces.

The selected inputs for each category were chosen based on the significance of extended producer responsibility (EPR) in e-waste management systems. The EPR strategy has been gaining considerable attention in recent years with respect to e-waste management. The EPR approach is a policy tool that places the responsibility and accountability of taking back consumers' obsolete electronic products onto producers [68].

### 3.1.1. The First Dataset

It was imperative to perform a document analysis to comprehend the two case studies, which was also valuable with respect to constructing a set of categories and indicators used to examine the effectiveness of e-waste infrastructures in both countries. As such, documents were imported into NVivo (version 13) to select, *filter, and examine* keywords and phrases related to e-waste management, as shown in Table A2 (Appendix B).

This strategy was sufficient for a deductive, contextual examination to obtain a broad and exhaustive set of publicly available material. NVivo allows us to code data related to selected text. In this context, we highlighted the related text and imported the text into its respective code. NVivo also allows for exportation of the resulting codes into a table format (CVS), as in Table A2 in Appendix B. The entire process was completed in four steps:

In step 1, the empirical data in PDF format (433 documents) were exported into the NVivo project.

In step 2, all documents were scanned for keywords and phrases related to government e-waste management. NVivo enables textual content analysis of other meaningful objects related to e-waste to be associated with codes (e.g., a picture, graph, banner, poster, etc.). In addition, NVivo allows for sentiment analysis for the codes associated with the following predefined classifications:

- *Attitude:* Governments' general approaches toward a favorable or unfavorable policy/issue related to e-waste matters;
- *Behaviors:* Include features such as actions, judgments, habits, values, beliefs, awareness, and perceptions attached to accepting, rejecting, or ignoring environmental motivation and costs associated with effectively managing e-waste as citizens, government officials, businesses, etc.;
- *Synonym:* Effect of a set of policies toward mobilizing resources;
- *Mixed*: Claims or statements that reflect attitudes that are neither clearly positive or negative;
- *Negative:* An opposing opinion or a disagreement where an opinion is not shared with someone or a group;
- *Positive:* Specifying positive opinions or agreement with other stakeholders or ideas; A shared opinion; and
- *Neutral:* These important features allow us to analyze text based on other environmental parameters, such as $CO_2$ emissions, costs associated with e-waste, resource mobilizations, etc.

In step 3, codes were explored using the sentiments described above.

In step 4, the results were exported into a .csv file, as shown in Table A2 in Appendix B.

After the document analysis, the next phase was to create a set of criteria and indicators to measure efficiency. Four categories were constructed, each with corresponding criteria and indicators. The four generic categories were clarity; responsiveness and inclusivity; data, reporting, evaluation, and compliance; and emissions. The corresponding criterion for clarity is well-defined scope. For responsiveness and inclusivity, the criterion is stakeholder participation. For data, reporting, evaluation, and compliance, there are four criteria: EPR, e-waste generation, e-waste collection and recycling, and e-waste disposal. Lastly, the criterion for $CO_2$ Emissions is $CO_2$/sector (kt/sector). Numerical values were assigned for each indicator in the Switzerland and Canada columns retrieved from these documents. As shown in Appendix B, a total of 32 indicators were constructed.

3.1.2. The Second Dataset

The second data source used in this study was associated with research papers available in the Scopus Database. Scopus has a reach set of published papers regarding sustainability and its pillars (economic, social, and environmental). As shown in Figure 1, the total of 463 research files were downloaded from the Scopus site, of which 344 (74.3%) were full research articles. Other files were conference papers (8%), reviews (7.8%), book chapters (3.2%), notes (2.4%), and books (1.3%), among others. The period of publication spans from 1972 to 2021, with 71.3% of documents published since 2012.

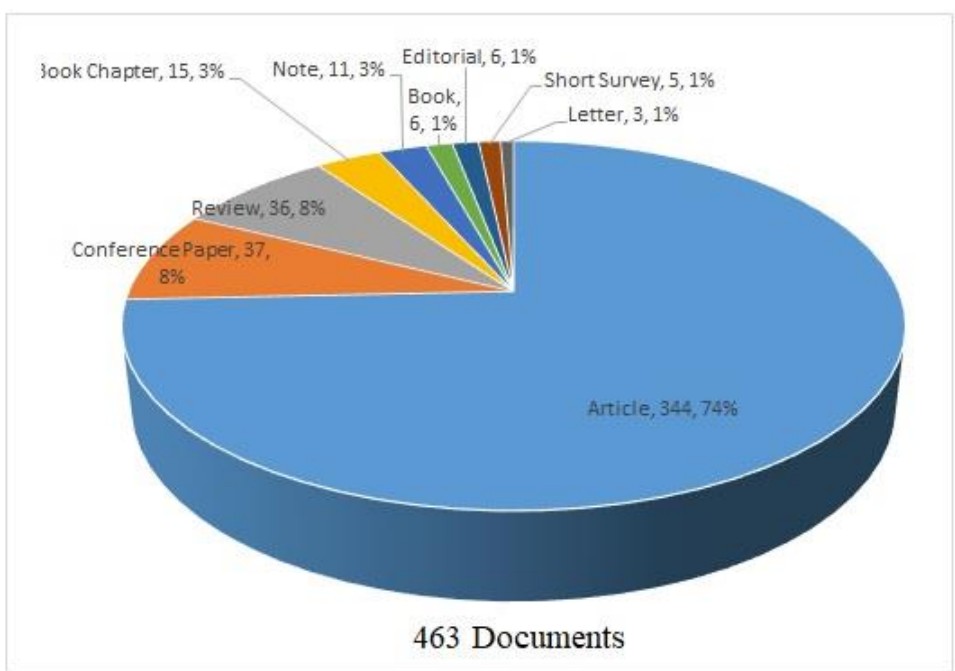

**Figure 1.** Types of research articles from the Scopus Database.

We used the following search query to extract data:

TITLE-ABS-KEY (("e-waste") OR ("e-waste management") OR ("regulation") OR ("total e-waste") OR ("environmental legislation") OR ("circular economy") OR ("control toxic") OR ("Smart City") OR ("Smart Cities") OR ("Climate Change") OR ("ICT") OR ("Adaptation") OR ("Environmental social commitment") OR ("stewardship") OR ("transparency") OR ("solidarity") OR ("corporate social responsibility") OR ("Leadership") OR ("extended producer responsibility") OR ("environmental tax") OR ("Compliance") OR ("e-waste generation") OR ("innovation") OR ("e-waste collection") OR ("e-waste collection rate") OR ("environmental health") OR ("transportation") OR ("ecological footprint") OR ("environmental footprint") OR ("waste") OR ("pollution") OR ("pollutant") OR ("recycling") OR ("circular economy") AND ("Canada") OR ("Canadian") AND ("Switzerland") OR ("Swiss")) AND PUBYEAR > 1972 AND PUBYEAR < 2022.

Moreover, we deployed a deep learning, unsupervised approach called the BERT model for text mining and analysis [69] of 463 PDF files constituting more than five thousand pages (23,028 pages in total). In the following section, we explain our text mining approach for this dataset.

Figure 2 shows the collected documents based on the subject area.

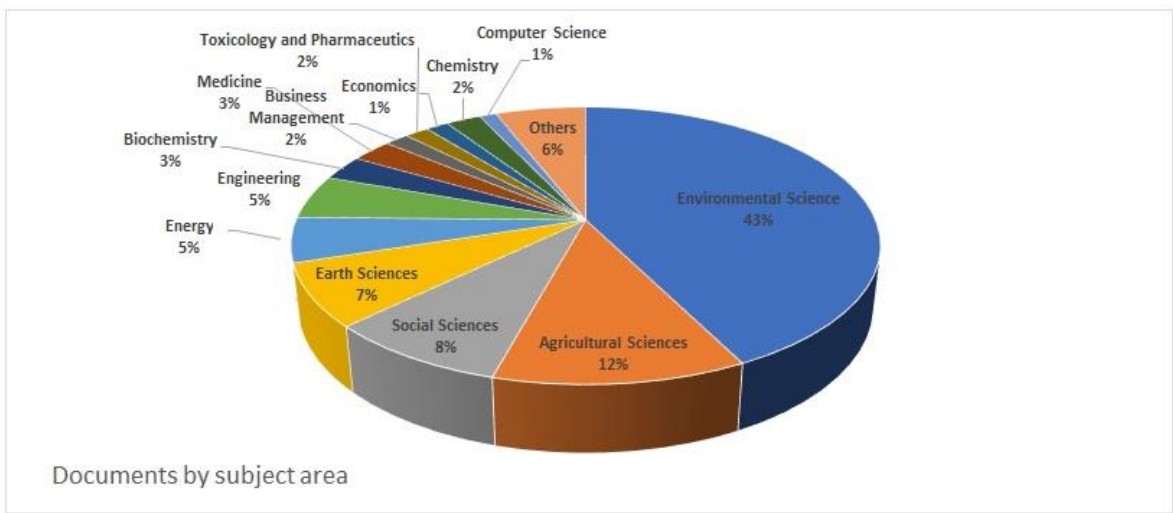

**Figure 2.** Documents by subject area from Scopus Database.

*3.2. BERT Text Analysis*

To classify words associated with the current project, we deployed a neural network deep learning technique for NLP called BERT. BERT (Bidirectional Encoder) is a transformer-based, bidirectional, unsupervised machine learning model in NLP. It is a pre-trained model that works from left to right and right to left. As BERT uses a transformer to obtain the meanings of words, it is a more attention-based algorithm than other traditional sequential text processing techniques. As such, it is used in many business applications, such as social media sentiment analysis, in chatbots to answer queries, predict text when writing emails, and summarize long legal documents [70], among other applications.

BERT performs like attention-based RNN (recurrent neural network) models for joint intent detection [71]; it builds a vector with an attention-based encoder–decoder or transformer capable of mapping sequences of varying lengths [72] concerning a keyword and its surrounding context. Moreover, BERT converts text into numbers, a helpful transformation necessary for making predictions using machine learning techniques.

Furthermore, unlike other machine learning applications, such as classification problems, metrics such as accuracy cannot be used to determine the system's effectiveness by computing the precision–recall curve. Instead, the mean average precision (MAP) metric is more helpful in quantifying the relevance of extracted keywords. This metric has embedded efficiency measures. As mentioned in [72], MAP corresponds to the area below the precision–recall curve and computed using Equation (1):

$$\text{MAP} = \frac{\sum_{n=1}^{|retrieved|} P_n \, r_n}{|relevant|} \tag{1}$$

where $P_n$ is the precision at the $n$ top-most returned results, and $r_n$ is a binary function indicating whether the $n$-th item in the returned ranked list is a relevant object (true positive) (p. 50).

As our dataset comprises a wide variety of research papers, we broadly classified them into four subdatasets, mapping the four categories on data with an approach similar to that applied to our first dataset (e.g., four text files). Then, we deployed a version of BERT called KeyBERT to measure the efficiency of keywords associated with Table A2 in Appendix B.

We trained our BERT model for text analysis using the BERT Tokenizer to convert the keywords into a vector of words. One of our goals in this study was to determine how to quantify whether an extracted keyword was relevant. Among the popular efficiency measures, namely the mean reciprocal rank (MRR) and mean average precision (MAP) metrics for quantifying the relevance of extracted keywords, we selected the MAP measure.

MAP is a prevalent scoring method used in information retrieval mainly when the user is expecting more than one possible relevant result for their search query. We performed the following steps to measure the efficiency the keywords in our dataset.

In step 1, we used the PyPDF2 library to convert PDF research files into text. Figures A1 and A2 in Appendix C show segments of the code used for this conversion. One of the challenges we had in this project was mapping the context of research works with the categories found in the first dataset, as shown in Table A2 in Appendix B. As discussed in the next section, it was necessary to split text documents into smaller corpora to map the main keywords associated with this study.

In step 2, we applied a relevant language model from the BERT transformers to our dataset files. In particular, keyBERT focuses mainly on the semantic context; a fixed-sized vector extracted from the dataset corresponds to these semantics.

In step 3, we applied natural language processing (NLP) techniques, such as CountVectorizer, to extract keywords and n-gram expressions from the dataset.

In step 4, the keywords extracted in step 2 were embedded into a new fixed-size vector using the same model as that used in step 1.

In step 5, the results of two types of embeddings, keyword embeddings and dataset/document embeddings, were considered to determine the most similar and extracted keywords among both the embeddings.

Finally, in step 6, we exported the keyword efficiency associated with our BERT analysis to a CSV file. The file contains 25 matching keywords associated with Table A2 in Appendix B. Data in this file were used as an output (efficiency), whereas data from the first dataset were used as input into our DEA analysis, as explained in the following section.

## 4. Data Envelopment Analysis and Results

DEA is a powerful and valuable service management and benchmarking tool created by Charnes et al. in 1978 [73] to assess non-profit and public sector organizations [74]. It is an efficiency rating technique for the evaluation of the relative efficiency of the units under investigation, the so-called decision-making unit (DMU). DMUs comprise multiple inputs and outputs, the evaluation results of which are derived from input and output data. DEA heavily depends on linear programming, making it a robust tool compared to other product management instruments.

In simple terms, efficiency is measured as an output-to-input ratio. The more outputs for every input unit, the greater the efficiency [74]. Once the highest output level is attained for every input, complete or optimum efficiency has been achieved; thus, it is impossible to achieve more efficiency without new technology or other enhancements in the production procedure [74].

As mentioned in [75] and as depicted in Equation (2), DEA aims to maximize $\theta$ (the efficiency score) for the DMUs under investigation. DEA benchmarks a DMU as an efficient unit if the value of $\theta$ is 100%. In other words, the practices with the highest efficiency have a rating of one, and less-efficient practices are rated as less than one.

$$\text{Maximize } \theta = \frac{\sum_{r=1}^{s} u_r y_{rj}}{\sum_{i=1}^{m} v_i x_{ij}} \tag{2}$$

s.t.

$$\sum_{r=1}^{s} u_r y_{rj} - \sum_{i=1}^{m} v_i x_{ij} \leq 0, \; j = 1, .., n$$

$$u_r \geq \varepsilon \, , \; v_i \geq \varepsilon \, \forall \, r, \, i$$

where:

$j$ = number of DMUs;

$\theta$ = the efficiency of a DMU;

$y_{rj}$ = the amount of output ($r$) used by DMU $j$;

$x_{ij}$ = the amount of input ($i$) used by DMU $j$;

$i$ = number of inputs used by DMUs;

$r$ = number of outputs generated by DMUs;
$u_r$ = coefficient or weight assigned by DEA to output $r$;
$v_i$ = coefficient or weight assigned by DEA to input $i$;

DEA compares service units, assuming that all resources and services are sufficient, and benchmarks the most efficient units or best practice units. Using the DEA model, an efficient frontier is built using the available data from all DMUs, as shown in Figure 3. Frontiers are those in which an efficiency limit is used to classify the different DMUs. The efficiency frontier is based on actual observations, and only the cases of best practices belong to it. We deployed a DEA program [74] associated with Stata Software version 15. The program offers efficiency options such as constant return to scale (CRS) [75] and variable returns to scale (VRS) [76].

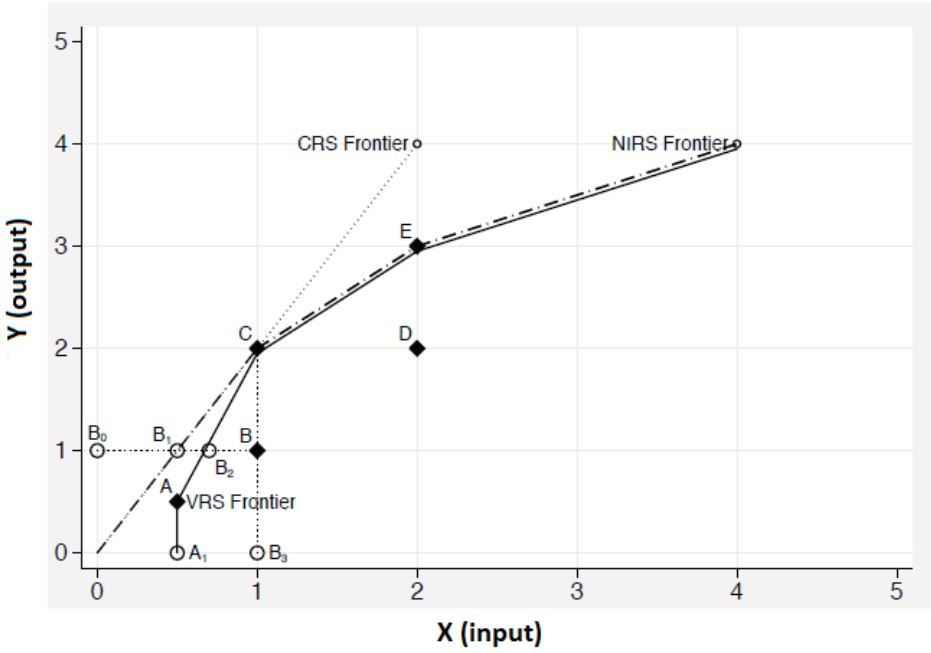

**Figure 3.** Concepts of efficiency and returns to scale [77].

As mentioned by Ji and Lee [76] frontiers determined by economies of scale are presented in Figure 3, using five DMUs labeled A through E. The CRS, VRS, and nonincreasing returns to scale frontiers are displayed. If CRS is assumed, the only efficient DMU would be C. However, DMUs A, C, and E are efficient if VRS is assumed. Therefore, we selected SRC and VRS returns to scale (IRS) methods for our analysis.

*DEA Efficiencies Results*

The DEA analysis measured the efficiency of the input and output values for each criterion mentioned earlier for Canada and Switzerland. We used the frequency analysis from the first dataset as input resource utilization for an optimal e-waste management approach generated from the second batch of research documents (output). The quantitative results are presented in Table 1. The theta column shows the efficiency results, with values ranging from zero to one, where one is the most efficient DMU located on the frontier line, indicating that the DMU is benchmarked. The results are input-oriented and based on the variable returns to scale (VRS) efficiency measurement, where VRS-TE is variable return to scale (same value as theta), and CRS_TE is a constant return to scale with DMUs operating at their optimal scale. The input-oriented method indicates that we are interested in the actual performance of e-waste management in these countries. The return to scale (RTS) column specifies variable returns to scale in the form of increasing return to scale (1) and decreasing (−1) or nonincreasing returns to scale (zero) (see [76] for more details). Moreover, the table shows the measurement model, breaking efficiency into technical (TE)

and scale efficiencies in DEA [76]. Finally, we selected the stage (2) that offers to an optimal reference by reducing the slack values.

**Table 1.** DEA VRS-INPUT-oriented efficiency results.

| DMU | Input | Output | Rank | Theta | CRS_TE | VRS_TE | SCALE | RTS |
|---|---|---|---|---|---|---|---|---|
| FLS-S | 7 | 0.671706 | 14 | 0.582298 | 0.533438 | 0.582298 | 0.916092 | 1.000000 |
| FLS-C | 4 | 0.457710 | 6 | 0.801966 | 0.636112 | 0.801966 | 0.793190 | 1.000000 |
| EWT-S | 7 | 0.995590 | 1 | 1.000000 | 0.790652 | 1.000000 | 0.790652 | −1.000000 |
| EWT-C | 2 | 0.160000 | 1 | 1.000000 | 0.444727 | 1.000000 | 0.444727 | 1.000000 |
| ACT-S | 7 | 0.935421 | 4 | 0.821225 | 0.742869 | 0.821225 | 0.904586 | −1.000000 |
| ACT-C | 7 | 0.777143 | 11 | 0.643409 | 0.617172 | 0.643409 | 0.959221 | 1.000000 |
| ESC-S | 7 | 0.814225 | 9 | 0.664902 | 0.646621 | 0.664902 | 0.972506 | 1.000000 |
| ESC-C | 7 | 0.388571 | 23 | 0.418194 | 0.308585 | 0.418194 | 0.737901 | 1.000000 |
| ENK-S | 7 | 0.519006 | 17 | 0.493793 | 0.412171 | 0.493793 | 0.834703 | 1.000000 |
| ENK-C | 7 | 0.480000 | 18 | 0.471186 | 0.381194 | 0.471186 | 0.809011 | 1.000000 |
| ADT-S | 7 | 0.807775 | 10 | 0.661163 | 0.641498 | 0.661163 | 0.970257 | 1.000000 |
| ADT-C | 7 | 0.565714 | 15 | 0.520865 | 0.449264 | 0.520865 | 0.862535 | 1.000000 |
| ACC-S | 7 | 0.228571 | 27 | 0.325458 | 0.181521 | 0.325458 | 0.557739 | 1.000000 |
| ACC-C | 7 | 0.189676 | 28 | 0.302914 | 0.150632 | 0.302914 | 0.497276 | 1.000000 |
| REG-S | 5 | 0.676458 | 5 | 0.819073 | 0.752097 | 0.819073 | 0.918230 | 1.000000 |
| REG-C | 5 | 0.205714 | 20 | 0.437094 | 0.228716 | 0.437094 | 0.523265 | 1.000000 |
| SOL-S | 5 | 0.234286 | 19 | 0.460278 | 0.260483 | 0.460278 | 0.565925 | 1.000000 |
| SOL-C | 5 | 0.197192 | 22 | 0.430179 | 0.219241 | 0.430179 | 0.509651 | 1.000000 |
| MAN-S | 5 | 0.899429 | 1 | 1.000000 | 1.000000 | 1.000000 | 1.000000 | 0.000000 |
| MAN-C | 5 | 0.650324 | 7 | 0.797867 | 0.723041 | 0.797867 | 0.906217 | 1.000000 |
| TRP-S | 77 | 0.494600 | 41 | 0.043604 | 0.035708 | 0.043604 | 0.818911 | 1.000000 |
| TRP-C | 85 | 0.360000 | 44 | 0.033076 | 0.023544 | 0.033076 | 0.711832 | 1.000000 |
| POP-S | 95 | 0.406048 | 45 | 0.031561 | 0.023761 | 0.031561 | 0.752854 | 1.000000 |
| POP-C | 79 | 0.322857 | 43 | 0.033680 | 0.022719 | 0.033680 | 0.674545 | 1.000000 |
| DAT-S | 10 | 0.494600 | 24 | 0.335753 | 0.274952 | 0.335753 | 0.818911 | 1.000000 |
| DAT-C | 10 | 0.400600 | 29 | 0.297616 | 0.222697 | 0.297616 | 0.748269 | 1.000000 |
| RCT-S | 10 | 0.993521 | 8 | 0.695697 | 0.552306 | 0.695697 | 0.793890 | −1.000000 |
| RCT-C | 10 | 0.117143 | 38 | 0.200000 | 0.065121 | 0.200000 | 0.325604 | 1.000000 |
| CMP-S | 10 | 0.485961 | 25 | 0.332248 | 0.270150 | 0.332248 | 0.813096 | 1.000000 |
| CMP-C | 10 | 0.240000 | 34 | 0.232457 | 0.133418 | 0.232457 | 0.573946 | 1.000000 |
| CMT-S | 10 | 0.741040 | 21 | 0.435739 | 0.411950 | 0.435739 | 0.945407 | 1.000000 |
| CMT-C | 10 | 0.234286 | 35 | 0.230139 | 0.130242 | 0.230139 | 0.565925 | 1.000000 |
| LDS-S | 10 | 0.371429 | 30 | 0.285781 | 0.206480 | 0.285781 | 0.722514 | 1.000000 |
| LDS-C | 10 | 0.146868 | 37 | 0.200000 | 0.081645 | 0.200000 | 0.408226 | 1.000000 |
| EWK-S | 201 | 0.862857 | 47 | 0.024137 | 0.023864 | 0.024137 | 0.988679 | 1.000000 |
| EWK-C | 757 | 0.818575 | 50 | 0.006172 | 0.006011 | 0.006172 | 0.974008 | 1.000000 |
| EWD-S | 123 | 0.725140 | 42 | 0.034901 | 0.032773 | 0.034901 | 0.939024 | 1.000000 |
| EWD-C | 101 | 0.416631 | 46 | 0.030111 | 0.022932 | 0.030111 | 0.761570 | 1.000000 |
| CO2-S | 4.57 | 0.349892 | 12 | 0.606220 | 0.425619 | 0.606220 | 0.702086 | 1.000000 |
| CO2-C | 15.69 | 0.102857 | 40 | 0.127470 | 0.036443 | 0.127470 | 0.285895 | 1.000000 |
| EWQ-S | 8.99 | 0.280000 | 32 | 0.276625 | 0.173142 | 0.276625 | 0.625906 | 1.000000 |
| EWQ-C | 226 | 0.222462 | 49 | 0.009971 | 0.005472 | 0.009971 | 0.548803 | 1.000000 |
| CSB-S | 7 | 0.234286 | 26 | 0.328770 | 0.186059 | 0.328770 | 0.565925 | 1.000000 |
| CSB-C | 7 | 0.125918 | 31 | 0.285714 | 0.099998 | 0.285714 | 0.349994 | 1.000000 |
| COT-S | 15.4 | 0.023760 | 39 | 0.129870 | 0.008577 | 0.129870 | 0.066042 | 1.000000 |
| COT-C | 173.8 | 0.257143 | 48 | 0.013775 | 0.008225 | 0.013775 | 0.597077 | 1.000000 |
| ECF-S | 10 | 0.168571 | 36 | 0.203477 | 0.093710 | 0.203477 | 0.460543 | 1.000000 |
| ECF-C | 10 | 0.276180 | 33 | 0.247136 | 0.153531 | 0.247136 | 0.621239 | 1.000000 |
| INN-S | 10 | 0.942860 | 13 | 0.590330 | 0.524144 | 0.590330 | 0.887883 | −1.000000 |
| INN-C | 10 | 0.900864 | 16 | 0.502985 | 0.500798 | 0.502985 | 0.995652 | −1.000000 |

Note: Options: RTS(VRS) ORT(IN) STAGE(2). DMU-S = Switzerland, -C = Canada. The return to scale (TRS) for VRS frontier (−1: decreasing, 0: nonincreasing, 1: increasing).

As mentioned in [76], the two-stage implementation reduces the slack and identifies the best optimal values. In addition, the slack issues in DEA models disappear as the number of DMUs increases because the DEA piecewise linear frontier becomes smoother, with less chance of running the Farrell point to the input or output axes (p. 270).

As depicted in Table 1, the results show that the DMU frontiers for Switzerland are EWT and MAN, whereas the frontier for Canada is EWT, as described below.

EWT is the total number of regulations and organizations at the federal level concerning the roles and responsibilities of stakeholders for the safe collection, return, and disposal of e-waste.

MAN represents the overall management of e-waste, highlighting the efficiency of policy implementation and the total number of organizations participating in e-waste services and facilitating cooperation between public and private sectors and citizens. Moreover, these organizations incorporate feedback from consumers and service providers to address e-waste management issues.

The relative efficiency analysis shows that Canada performed better in areas such as FLS (toxic control and prohibiting illegal exports) and ECF (economic flow) than Switzerland, whereas Switzerland was relatively efficient in the area of e-waste regulations.

Moreover, both countries are relatively efficient (almost to the same degree) in categories such as DAT, SOL, and INN.

DAT refers to the implementation of data analytical tools [77,78] used by producer responsibility organizations (PROs), such as collection and/or take-back and collection targets, reuse and recycling targets, technical product standards, e-waste volume reduction targets, encouraging repair and durability for reuse of products, resource recovery, etc.

SOL refers to solidarity, stressing that the development of sustainable electronic waste policies can only be obtained through the interplay of social, economic, and environmental sustainability, as well as cooperation, unity, and solidarity between the formal and informal sectors of the waste management industry. Moreover, as outlined in the UN's SDGs, cooperation is required at domestic and global scales.

INN represents innovation—not only in terms of recycling e-waste but also extracting and reusing e-waste materials, in addition to innovative methods for manufacturing eco-friendly EEE to reduce the amount of e-waste.

The least efficient DMUs related to Canada are associated with $CO_2$ emission; in particular, they show a high level of emissions generated per capita in the transportation section. Additionally, EWK (e-waste generated in kilo tons) for Canada and Switzerland shows an alarming value, indicating that more resources need to be allocated to meet the demand.

## 5. Discussion

There is no doubt that Switzerland has been an inspiration for many countries. The Swiss system is a well-established model that offers the complete take-back of e-waste; recycling systems in the country are financed through advance recycling fees (ARF), which are catered to recyclers, retailers, distributors, and customers. Although the Swiss e-waste management system has been a benchmark for many countries, their government and industry consensus has revealed that the country's present take-back and recycling systems can be enhanced to be more effective, equitable, and financially feasible.

Canada needs to learn from achievements and shortcomings of the Swiss model rather than aiming to *mimic* every aspect of the system because, contextually, what works in Switzerland may not work in Canada. Provinces in Canada, such as British Columbia, have established advanced recycling and collection facilities, whereas others have only depot locations or none at all. Additionally, there is a lack of demographic data in Canada, which is a major indicator of recycling rates. For instance, provinces with higher densities, immigrant populations, and residential/commercial buildings will differ from provinces with lower densities characterized by rural communities in terms of recycling rates. On a similar note, nations with successful e-waste EPR programs are only effective because

they have data on the points mentioned earlier and have worked toward controlling for inadequacies to improve their systems. Canada needs to take a similar approach by filling in gaps in data and devising steps to communicate and control for such variations to create sustainable e-waste management systems across the nation. Therefore, in this section, we will discuss some policy considerations for Canada to achieve this goal in the future.

Providing incentives to enhance environmental performance is associated with the degree to which manufacturers are legally accountable for compliance. For example, the fundamental principles of economics demonstrate that EPR programs often function best when individual manufacturers are responsible for the waste their products create. Ultimately, this provides a direct pricing indicator such that manufacturers have a clear incentive to enhance environmental performance.

Although provincial EPR policies in Canada have attempted to allocate legal responsibilities toward manufacturers, it is unknown whether producers meet these standards in reality because a majority transfer such responsibilities to their PROs.

The recent amendments to EPR laws in Ontario highlight this problem. For example, Ontario is Canada's first province to combine the fundamentals of individual producer responsibility into its EPR policies. Under this regulation, the responsibility to collect WEEE lies with the producer, not the PRO, if the producer decides to join one. For instance, once a manufacturer collaborates with a PRO to handle its WEEE, the manufacturer is still theoretically obligated if the PRO fails to achieve its goals. However, because the regulation is still in its early stages of enforcement, it is too early to investigate the effectiveness of these programs across Ontario and other provinces.

As mentioned earlier, one of the reasons for a poor score of e-waste management in Canada is fragmented cooperation between provinces. Whereas greater coordination and collaboration of policies across provincial jurisdictions the management of WEEE can lead to many advantages, complete harmonization is a challenge that all provinces must overcome. Under present laws, provincial EPR programs related to e-waste cover a wide range of products and materials, and manufacturers must adhere to the legislation in their region. Therefore, it is time for Canada to rethink its dominant way of framing e-waste flows. Finally, a unified approach should be developed that will reduce costs by coordinating education and outreach, behavioral issues, and legal blueprints throughout the country [79].

Although the possibility of complete harmonization among provinces seems unlikely, it is not impossible once a few obstacles are conquered. First, the federal government is entirely restricted in its scope to enforce national EPR programs. A good start for the federal government, specifically ECCC (Environment and Climate Change Canada), would be to play a larger role in harmonization procedures, as well as by strengthening legislation and ratifying the Ban Amendment to prohibit transboundary movements of e-waste. Nevertheless, it remains to be seen how influential the federal government could be. The federal government can achieve these goals by:

- Expanding on e-waste management planning beyond municipalities;
- Expanding and implementing *Phase 2* of the Canada-wide Action Plan;
- Remedying the lack of federal and provincial targets and goals by focusing on ways to reduce e-waste flows and consumption;
- Exploring the creation of a federal reporting protocol to gather data from waste audits on a yearly or continuous basis to enable effective collection and data analysis; and
- Calling on provinces to compare e-waste and waste characterization data with waste policies and other e-waste reduction measures and initiatives, ensuring that they align with proposed climate targets.

## 6. Conclusions

In this study, we attempted to provide a snapshot of the e-waste management systems of Canada and Switzerland by critically examining and discussing Canadian and Swiss e-waste management strategies with a focus on each country's legal framework,

waste collection/recycling/disposal practices, system financing, producer responsibility, and compliance practices. Through a mixed efficiency approach, the research design for this sustainable e-waste management study stems from the most effective techniques for analysis of both nations through a comparative policy [80] lens.

After examining both countries in considerable detail, we conclude that extensive dependent and independent factors result in countries selecting various regulations and strategies for management of e-waste flows in specific ways. To the best of our knowledge, this is the first study to incorporate DEA and NLP-based BERT analysis to identify the most efficient policy deployment concerning e-waste management.

In this study, we raised three research questions (RQs). RQ1 indicates the challenges and opportunities in Canada's current e-waste management system. Our results revealed that Canada performed well in EWT or e-waste regulation in terms of stakeholders' roles and responsibilities for the safe collection, return, and disposal of e-waste. Canada was also relatively efficient in international collaboration for e-waste management (variable SOL), deploying innovation with respect to data collection and analytics (DAT) and technical innovation (INN) concerning e-waste.

RQ2 was related to policy measures that can be used to improve Canadian e-waste. As shown in Table 1, Canada must deploy more resources in response to the total amount of e-waste generated (EWK) in this country; this means installing more collection areas across cities and processing collected e-waste using more treatment facilities.

Finally, RQ3 raised the question of how Canada can learn lessened from the Swiss e-waste management system. As shown in Table 1, the Swiss system outperformed Canada in terms of overall management of e-waste (MAN), which means that Canada needs to be more proactive in terms of the efficiency of policy implementation and the total number of organizations participating in e-waste services. Moreover, cooperation between the public and private sectors and citizens is critical as the amount of e-waste continues to increase rapidly. In this context, Canada needs to implement a national communication strategy to address the importance of shifting upwards on the waste hierarchy and help distinguish between e-waste management and prevention goals. Finally, the country is currently too focused on recycling materials, and greater emphasis needs to be placed on reducing and reusing, repair initiatives, landfill bans, and energy input restrictions.

We believe that this study provides valuable knowledge and insights for all stakeholders involved in managerial decisions, as well as information with respect to how legislation and practices are constructed and enforced. E-waste management will continue to evolve in an era threatened by climate change and its detrimental effects. As the amount of e-waste continues to rapidly increase, we hope that discussions about efficient and sustainable management techniques will gravitate toward a central focus on how stakeholders visualize and perceive the risks of buying new electronic and electrical equipment for short periods. Finally, we hope that communication with respect to how to correctly dispose of e-waste becomes more convenient and accessible for individuals so that the decisions and procedures for e-waste management become less complex.

**Author Contributions:** Formal analysis, S.A.; Methodology, F.S. All authors have read and agreed to the published version of the manuscript.

**Funding:** This research received no external funding.

**Conflicts of Interest:** The authors declare no conflict of interest.

# Appendix A

**Table A1.** Swiss and Canadian e-waste management features.

| Feature | Switzerland | Canada |
|---|---|---|
| Accountability of WEEE Collection and Treatment | • Responsibility of all producers, distributors, retailers, recyclers, and consumers (financing system through ARFs)<br>• No distinction among product categories | • Responsibility of all producers, distributors, retailers, recyclers, and consumers (financing system through EHFs)<br>• A total of nine WEEE groups, irrespective of customer, commercial, or residential purposes |
| Manufacturers | • SWICO<br>• SENS<br>• SLRS<br>• INOBAT | • EPSC<br>• EPRA |
| Recyclers | • SWICO<br>• SENS<br>• SLRS<br>• INOBAT | • Daily activities certified through EPRA and EPSC RQP standards<br>• Call2Recycle |
| Federal Legislation | Yes<br>*ORDEA*<br><br>- At the forefront of providing the legal blueprint for e-waste management<br>- Section 2 outlines the rules for returning, taking back, and the disposal of electronic and electrical equipment<br>- *Article 3* summarizes duties for users for safe return of WEEE, as well as the responsibilities of producers and traders for taking back WEEE mentioned in *Article 4*<br>- Section 3 provides a strict framework of conditions that must be fulfilled before any sort of WEEE is allowed to be shipped externally for disposal purposes<br><br>• DETEC<br>• FOEN<br>• Basel Convention on Control of Transboundary Movements of Hazardous Wastes and their Disposal<br>• Basel Ban Amendment | No<br>*ECCC*<br><br>- Responsible for policies related to disposal, handling, and shipment of toxic waste and obliged to create official policy documents on MSW management in the country<br>- Records global agreements into national law<br><br>• CEPA (1999)<br>• PCB Waste Export Regulations under CEPA (1999)<br>• TDG Regulations<br>• CCME's community-based route is directly placed upon the waste hierarchy and incorporates values associated with reuse. The organization has also introduced a promising EPR model emphasizing that producers, brand owners, and importers are obligated to manage WEEE |
| National/Provincial/Territorial Legislation | Yes | • With the exception of Nunavut, all provinces and territories in Canada have implemented regulated EPR programs for management of e-waste<br>• Control and license provincial WEEE producers, recyclers, and treatment facilities |
| Municipal Legislation | • Appropriate balance of government involvement coupled with public–private partnerships<br>• Collaborations between PROs and Swiss cantons have been shown to reduce monopolies, and the introduction of competition considerably reduces management costs | • Responsibility for overlooking domestic waste management activities, specifically giving instructions related to recycling and disposal<br>• Can enforce domestic landfill bans<br>• Public education and outreach on recycling and disposal |
| Certifications (audits, verifications, approval procedures, compliance protocols, etc.) | • Audit system increased since introduction of the 50,625 standard<br>• Main adjustments include document analyses before the actual tour of the plants, and some checkpoints are not required to be inspected yearly. This allows for thorough inspection of the plant. | • Formal licensed agreements for recycling and processing facilities are received through audit and approval by the EPSC's national RQP, which outlines the minimum requirements for such organizations in terms of functioning in a safe manner under their provincial electronics stewardship programs<br>• R2 Standard, ERRP |

Note: **EHF:** environmental handling fee; **ARF:** advance recycling fee; **EPSC:** Electronic Product Stewardship Canada; **EPRA:** Electronic Products Recycling Association; **RQP:** Recycler Qualification Program; **SWICO:** Swiss Association for Information, Communication and Organizational Technology; **SENS:** Swiss eRecycling body for recycling of household appliances; **SLRS:** Recycling Guarantee, Swiss Foundation for Waste Management, Swiss Light Recycling Foundation; **INOBAT**: *Swiss federal collection system for used cells and batteries implemented in collaboration*; **ORDEA:** Ordinance on the return; **ECCC:** Environment and Climate Change Canada; **CEPA:** Canadian Environmental Protection Act; **PCB:** polychlorinated biphenyls; **TDG:** transportation of dangerous goods; CCME: Canadian Council of Ministers of the Environment; **DETEC:** Federal Department of the Environment, Transport, Energy and Communications; **FOEN:** Swiss Federal Office for the Environment; **ERRP:** Electronics Reuse and Refurbishing Program.

## Appendix B

**Table A2.** Criteria and indicators.

| Category | NVivo Code | Criteria | Indicators (Inputs) | Switzerland | Canada |
|---|---|---|---|---|---|
| **(A) Clarity** | GLS | Well-defined scope | Total number of governmental organizations and/or legislation for management of environmental issues at the federal level, including e-waste | 3 | 2 |
| | FLS | | Total number of regulations and/or organizations at the federal level for control of toxic substances and prohibition of illegal exports to developing countries | 7 | 4 |
| | EWT | E-waste | Total number of regulations and/or organizations at the federal level with respect to roles and responsibilities of stakeholders for safe collection, return, and disposal of e-waste | 7 | 2 |
| | PRO | | Total number of major nationwide PROs | 4 | 2 |
| **(B) Responsiveness and Inclusivity** | ACT MOT: Motivation ENC: Environmental costs NOM: Norms INT: Interactions ACC: Acceptance RSO: Reuse options CON: Constraints COV: Convenience RAT: Rationality | Stakeholder participation | The number and extent to which actors within the e-waste management services are involved in the planning, education, awareness, application, and assessment of those amenities (actors include producers, distributors and retailers, government, PROs, and citizens) | 7 | 7 |
| **(C)** | ADT | | Adapting to climate change through local municipal planning and promoting local adaptation policy development | 7 | 7 |
| **(D)** | ESC | | ESC: environmental and social commitment | 7 | 7 |
| **(E)** | ENK | | Environmental knowledge | 7 | 7 |
| | MAN | | Total number of organizations that motivate participation and facilitate partnerships for e-waste services between public and private sectors and incorporate feedback between consumers and service providers to address e-waste management issues | 5 | 5 |
| | REG | Regulation | E-waste regulations | 5 | 5 |
| | TRP | Transparency | Transparency in communicating with stakeholders | 5 | 5 |
| | POP | | Percentage of population aware of e-waste recycling programs in their locality | 95 | 79 |
| | SOL | | Solidarity and international engagement in global e-waste | 5 | 5 |
| **(F) Data, Reporting, Evaluation, and Compliance** | DAT | Extended producer responsibility (EPR) | Implementation of *data analytical tools* used by PROs, such as collection and/or take-back and collection targets, reuse and recycling targets, product technical standards, e-waste volume reduction targets, encouraging repair and durability for reuse of products, resource recovery, etc. | 10 | 10 |
| | LDS | Leadership | Leadership in e-waste implementation | 10 | 10 |
| | RCT | | Recycling targets | 10 | 10 |
| | TAX | | Implementation of *economic tools* by PROs, such as eco-fees, ARFs, material taxes/subsidies, etc. | 1 | 1 |
| | CMP | | Compliance | 10 | 10 |
| | EWK | E-waste generation | E-waste generated (kt) | 201 | 757 |
| | CMT | Commitment | Showcases the commitment toward a long-term waste strategy, as well as engaging stakeholders | 10 | 10 |
| | INN | Innovation | Deploying innovation for data collection and technical innovation | 10 | 10 |
| | ECF | Circular economy | Economic flow | 10 | 10 |
| | EWQ | | Small equipment | 8.99 | 226 |
| | EWS | | Small IT | 2.59 | 49.5 |
| | EPC | | Total e-waste placed on market (kg/capita) | 24.2 | 23.8 |
| | EWR | E-waste collection and recycling | Total number of collection points | 1300 | 2500 |
| | EWD | | E-waste documented to be formally collected and recycled (kt) [1] | 123 | 101 |
| | WRT | | E-waste collection rate (formal collection divided by e-waste produced) (%) | 67 | 20 |
| | WDS | E-waste disposal | Total quantity of electronics disposed (kt) | 127 | 375 |
| **(G) Emissions** | CO$^2$ | CO$^2$/sector (kt/sector) | CO$^2$/PC (kiloton metric) | 4.57 | 15.69 |
| | COT | | Transport | 15.4 | 173.8 |

[1] Sources: Data obtained from [11,41], SENS, SWICO, SLRS Technical Report 2020 (https://adobeindd.com/view/publications/0871633d-f33d-4a34-9312-7df241e5e9e2/tc49/publication-web-resources/pdf/201005_SE_Fachbericht_2020_EN.pdf (accessed on 18 February 2021). The Quality of Government (QoG) Institute, University of Gothenburg, Sweden (https://www.gu.se/en/quality-government (accessed on 18 February 2021) and The Environmental Performance Index (EPI), Yale University (https://epi.yale.edu/ (accessed on 18 February 2021).

**Appendix C**

```
def merger(output_path, input_paths):
    pdf_writer = PdfFileWriter()
    for path in input_paths:
        pdf_reader = PdfFileReader(path)
        for page in range(pdf_reader.getNumPages()):
            pdf_writer.addPage(pdf_reader.getPage(page))
    with open(output_path, 'wb') as fh:
        pdf_writer.write(fh)
if __name__ == '__main__':
    paths = glob.glob('samples/*.pdf')
    paths.sort()
    merger('output/pdf_merged.pdf', paths)
....

def pdf_splitter(path):
    fname = os.path.splitext(os.path.basename(path))[0]
    pdf = PdfFileReader(path)
    for page in range(pdf.getNumPages()):
        pdf_writer = PdfFileWriter()
        pdf_writer.addPage(pdf.getPage(page))

        output_filename = 'split/{}.pdf'.format(page+1)
......

if __name__ == "__main__":

 clear_text()
 fname = os.listdir('split/')
#fname : List contain pdf documents names.
#fname: must be sorted.
 fname.sort(key=lambda f: int(re.sub('\D', '', f)))
 length = len(fname)

 for i in range(length): #Repeat each operation for each document.

    text_output = pdf_to_text(('split/{}').format(fname[i]))
#Extract text with PDF_to_text Function call
    text1_output = text_output.decode("utf-8")
#Decode result from bytes to text
#Save extracted text to TEXT_FILE
    with open("output/Output.txt", "a", encoding="utf-8") as text_file:
      text_file.writelines(text1_output)
```

**Figure A1.** Snapshot of text files generated from PDF for the second dataset.

```
from keybert import KeyBERT
from nltk.tokenize import sent_tokenize, word_tokenize
from nltk.corpus import stopwords
…
Topics = {'management in clarity': [], 'mangement in responsiveness ': [],
'management in compliance ': [], 'management in emissions: []}
…
tokens = word_tokenize(data)
tokens_sw = [word for word in tokens if not word in stopwords
data = (" ").join(tokens_sw)
k_model = KeyBERT()
keywords = k_model.extract_keywords(data, keyphrase_ngram_range=(2,top_n = 30)
…
if (len(precision) == 0):
if k == 'Keywords for management':
avg_pre_man_s.append(0)
elif g == 'Keywords for management':
MAP_man_s = sum(avg_pre_man_s)/len(avg_pre_man_s)

MAP_man_s: 0.8994290420345612
```

**Figure A2.** A Snapshot of code generation for the second dataset.

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
