# Peer review of "A Transformer-Based Machine Learning Approach for Sustainable E-Waste Management: A Comparative Policy Analysis between the Swiss and Canadian Systems"

_sustainability, doi:10.3390/su142013220_

Round 1
Reviewer 1 Report
Please find the attachment.

Author Response
Comment 1:
Section 2.4 highlights as ‘E-Waste Treatment Methods’. The section does not clearly mention what is the priority as ‘treatment’ or ‘recovery of value added elements. The authors rightly state that ‘There is no doubt that e-waste holds great financial value due to the amounts of substances found within each piece of WEEE (Waste Electrical and Electronic Equipment), such as silver, gold, platinum, palladium, etc. However, a more relevant information is expected regarding the same. Various electronic products contain variable amounts of base and precious metals. A detailed list of elements have been provided as per reports of Gautam et al 2022 (Pushpa Gautam, Chhail K. Behera, Indrajit Sinha, Gospodinka Gicheva, Kamalesh K. Singh, High added-value materials recovery using electronic scrap-transforming waste to valuable products, Journal of Cleaner Production, 330, 2022,129836). Some analyses from this report could be included.
Response 1:
Thank you for valuable information, we added the above paper in revised paper and added some relevant content from the above paper as suggested.
Gautam el al. [87], provide a detail list of valuable base and precious metals that can be extracted from electronic waste and be used in a wide range of industries. As the authors mentioned, a wide range of value-added products such as, high purity base and precious metals, metal and metal oxide nanoparticles, nanostructured alloys, nanocomposites, microparticles, and composites, has been recovered from the retrieved, low-cost, ubiquitous electronic scrap [87].
Comment 2:
The authors have rightly mentioned that ‘Some important metals in cellphones include copper and tin, as well as unique metals like lithium, cobalt, antimony, silver, gold, palladium etc. Specifically, 23 percent of the weight of a typical cellphone consists of valuable metals, with the rest being made of plastic and ceramic’. Recovery of these metals play a vital role and could be highlighted as the article is aimed to be published in ‘Sustainability’. As per the recent reports of Sahu et al., 2022, recovery of ‘energy critical elements’ from waste electrical and electronic equipment (WEEE) is a relevant information to be considered. The energy-critical elements include:
Response 2: Thank you for the valuable information, we have used some of the references mentioned by our respected reviewer, in the revised paper. Including Sahu, S., Mohapatra, M., & Devi, N. (2022), Gautam, P. et al. (2022); Jaffe, et al. (2011), Isidar yet al. (2019).
Reviewer 2 Report
The document has a good structure. However, some details need to be considered, as follows:
The introduction needs to be condensed.
The related works section defines several concepts, however, it doesn't present some related works for Canadian policies or Switzerland policies, or comparative policy analysis.
Line 62: When you say "electronic waste were produced nationally", it is not clear in which country or if it is worldwide and average by country.
Line 117: "of this thesis"
Line 135: "this thesis"
Switzerland was chosen as the most suitable country for comparison because environmental sustainability problems are an important aspect for the Swiss government and its residents. However, has Switzerland some similar characteristics to Canada? To facilitate the introduction of switzerland's policies in Canada.
No data showing the effectiveness of the Switzerland system is presented. It only presents that in 2018 Switzerland was ranked first on the Environmental Sustainability Index.
Section 1.2 is redundant. It repeats that academic literature has only focused on electronic waste policies in European nations and not in countries like Canada. Please reformulate this section.
Lines 174-175: "This principle is comprised in the following 174 three core facts." It seems something is missing in the paragraph.
Line 192: "Economic Sustainability" Some text is missing.
The methodology introduction describes several ways to assess policy. However, it is not clear which is selected or preferred for the study.
Line 428: "As shown in Figure 1, we tried to limit the number of articles to match the primary data set."
How do you try to limit the number of articles?
Which keywords did you use to search in Scopus?
Which is the time frame for the search?
Please create your own figures for Figure 1 and 2.
Section 3.2 It is not clear if you are presenting a new methodology or a methodology presented elsewhere. Furthermore, the objective of applying this methodology is not clear.
DEA is presented in section 4, with results. However, its methodology is not presented in section 3.
Furthermore, the objective to use DEA is not clear.
Figure 3 needs to be improved.
No discussion related to the results presented in table 1 is found.
Most of the comparative data is in the appendix, it is recommended to create a Figure to compare and include it in the text.
Author Response
Comment 1:
The document has a good structure. However, some details need to be considered, as follows:
The introduction needs to be condensed.
Response 1: Thank you for the comment; we have condensed the introduction section by removing some text from section 1.2 (lines 138-150) and combining the section with section 1.1.
The related works section defines several concepts; however, it doesn't present some related works for Canadian policies or Switzerland policies, or comparative policy analysis.
Comment 2:
Line 62: When you say "electronic waste were produced nationally", it is not clear in which country or if it is worldwide and average by country.
Response 2: Thank you for the comment; the issue has been addressed in the revised.
Comment 3:
Line 117: "of this thesis"
Line 135: "this thesis"
Switzerland was chosen as the most suitable country for comparison because environmental sustainability problems are an important aspect for the Swiss government and its residents. However, has Switzerland some similar characteristics to Canada? To facilitate the introduction of Switzerland’s policies in Canada.
No data showing the effectiveness of the Switzerland system is presented. It only presents that in 2018 Switzerland was ranked first on the Environmental Sustainability Index.
Response 2: Thank you for the comment; we have added some fresh data for Switzerland in the revised version to highlight its importance for Canada.
Comment 3:
Section 1.2 is redundant. It repeats that academic literature has only focused on electronic waste policies in European nations and not in countries like Canada. Please reformulate this section.
Response 3: Thank you for the comment. Please see our response to comment one above.
Comment 4:
Lines 174-175: "This principle is comprised in the following 174 three core facts." It seems something is missing in the paragraph.
Response 4: Thank you for the comment. The issue has been addressed in the revise.
Comment 5:
Line 192: "Economic Sustainability" Some text is missing.
Response 5: Thank you for your attention. Line 192 supposed to be a sub-heading. We fixed the issue.
Comment 6:
The methodology introduction describes several ways to assess policy. However, it is not clear which is selected or preferred for the study.
Response 6: Thank you for the comment, in the methodology section of the revised paper, we have mentioned our preferred method.
Comment 7: Line 428: "As shown in Figure 1, we tried to limit the number of articles to match the primary data set."
How do you try to limit the number of articles?
Response 7: Thank you for the comment, the limitation was based on the time. We have explained this in the revise.
Comment 8:
Which keywords did you use to search in Scopus?
Which is the time frame for the search?
Response 8: Thank you for the comment; we explained the period and the query syntax in the revised.
Comment 9: Please create your own figures for Figure 1 and 2.
Response 9: Thank you for the comment. Figure were generated as suggested.
Comment 10:
Section 3.2 It is not clear if you are presenting a new methodology or a methodology presented elsewhere. Furthermore, the objective of applying this methodology is not clear.
DEA is presented in section 4, with results. However, its methodology is not presented in section 3.
Furthermore, the objective to use DEA is not clear.
Response 10: In revised, we have clearly explained the importance of DEA and how the mathematical model has setup.
Comment 11:
Figure 3 needs to be improved.
Response 11: Thank you for the comment, we tried our best to provide a clearer figure.
Comment 12:
No discussion related to the results presented in table 1 is found.
Most of the comparative data is in the appendix, it is recommended to create a Figure to compare and include it in the text.
Response 12: We have explained the results and their meaning as depicted in table 1, in section 4.1.
Reviewer 3 Report
The paper is quite interesting in this field.
But there is a lack of the following points:
1. There is a lack of mathematical or logical parts.
2. Some statistical tests should be executed to verify the performance differences among algorithms.
3. The novelty of the proposed method should be provided clearly.
4. The abstract should be improved to show the contributions clearly.
5. Add a flow chart of the proposed method if possible.
Author Response
Comment 1: There is a lack of mathematical or logical parts.
Response 1: Thank you for the comment, in revised we included the mathematical model of our main DEA model.
Comment 2: Some statistical tests should be executed to verify the performance differences among algorithms.
Response 2:
Thank you for the comment. In response to our reviewer’s comment we added the following text to clarify performance measure computed by BERT.
Unlike other machine learning applications, like classification problems, one couldn't use metrics such as accuracy to determine the system's effectiveness by computing the Precision-Recall curve. Instead, the Mean Average Precision (MAP) metric is more helpful in quantifying how relevant these extracted keywords were. This metric has embedded efficiency measures. As mentioned by [93], MAP corresponds to the area below the Precision-Recall curve, computed using equation (2).
Response 2: Thank you for the comment; the accuracy of our Deep Learning model has been explained in detail in the revised paper.
Comment 3: The novelty of the proposed method should be provided clearly.
Response 3: Thank you for the comment, in DEA section we have included the importance of using DEA for evaluating the efficiency of metrics used in this study. We added the following text from Sherman and Zhu (2006).
Response 3: Thank you for the comment; we explained the DEA as an effective benchmarking technique in the revised paper.
Comment 4: The abstract should be improved to show the contributions clearly.
Response 4: We agree with our reviewer; the abstract section has been modified accordingly.
Comment 5: Add a flow chart of the proposed method if possible.
Response 5: In detail, we explained step-by-step keyword extraction Key BERT process in text.
Round 2
Reviewer 2 Report
The document still has some lines showing that this is a thesis work, not an article, like in line 119.
The discussion section or conclusions must present the response to the research questions from section 1.1
In the same vein, related works it more like an introduction. Because it is not related even to e-waste. It is a general definition of sustainability that can be eliminated. So, this section needs to be improved.
E-Waste treatment methods are not mentioned in the discussion section or results, then if this information is not relevant for the research, it can be eliminated from the related works. Or presented in a resumed paragraph.
Author Response
Comment 1:
The document still has some lines showing that this is a thesis work, not an article, like in line 119.
Response 1: Thank you for the comment the issue was fixed.
Comment 2:
The discussion section or conclusions must present the response to the research questions from section 1.1
Response 2: Thank you for the comment the research questions related to the results were discussed in conclusion section.
Comment 3:
In the same vein, related works it more like an introduction. Because it is not related even to e-waste. It is a general definition of sustainability that can be eliminated. So, this section needs to be improved.
Response 3: Thank you for the comment, the subsections related to sustainability in the introduction section have been removed.
Comment 4:
E-Waste treatment methods are not mentioned in the discussion section or results, then if this information is not relevant for the research, it can be eliminated from the related works. Or presented in a resumed paragraph.
Response 4:
Thank you for the comment, this section is related to research question 2, as we updated in the conclusion section.
Reviewer 3 Report
The paper has a good impact on the application.
Add one more numerical experiment if possible with sensitivity analysis.
Author Response
Comment 1:
The paper has a good impact on the application.
Response 1: Thank you for the comment, be appreciate it.
Comment 2:
Add one more numerical experiment if possible with sensitivity analysis.
Response 2:
The DEA version we used has a two-stage option; this option will reset the slack and sensitivity to zero. We have provided a few lines explaining why we used the 2-stage option. Finally, we used the author's (Ji & Lee, [78]) explanation to support our argument.